# The Toxic Effect of Herbicidal Ionic Liquids on Biogas-Producing Microbial Community

**DOI:** 10.3390/ijerph16060916

**Published:** 2019-03-14

**Authors:** Jakub Czarny, Agnieszka Piotrowska-Cyplik, Andrzej Lewicki, Agnieszka Zgoła-Grześkowiak, Łukasz Wolko, Natalia Galant, Anna Syguda, Paweł Cyplik

**Affiliations:** 1Institute of Forensic Genetics, Al. Mickiewicza 3/4, 85-071 Bydgoszcz, Poland; nat.gal91@gmail.com; 2Institute of Food Technology of Plant Origin, Poznan University of Life Sciences, Wojska Polskiego 31, 60-624 Poznan, Poland; agnieszka.piotrowska-cyplik@up.poznan.pl; 3Institute of Biosystem Engineering, Poznan University of Life Sciences, 50 Wojska Polskiego St., 60-637 Poznan, Poland; andrzej.lewicki@up.poznan.pl; 4Faculty of Chemical Technology, Poznan University of Technology, Berdychowo 4, 60-965 Poznan, Poland; agnieszka.zgola-grzeskowiak@put.poznan.pl (A.Z.-G.); anna.syguda@put.poznan.pl (A.S.); 5Department of Biochemistry and Biotechnology, Poznan University of Life Sciences, Dojazd 11, 60-632 Poznan, Poland; lukasz.wolko@up.poznan.pl; 6Department of Biotechnology and Food Microbiology, Poznan University of Life Sciences, Wojska Polskiego 48, 60-627 Poznan, Poland; pcyplik@wp.pl

**Keywords:** anaerobic digester, biodegradation, herbicidal ionic liquids, MiSeq technology

## Abstract

The aim of the study was to evaluate the effect of herbicidal ionic liquids on the population changes of microorganisms used in a batch anaerobic digester. The influence of the following ionic liquids: benzalkonium (2,4-dichlorophenoxy)acetate (BA)(2,4-D), benzalkonium (4-chloro-2-methylphenoxy)acetate (BA)(MCPA), didecyldimethylammonium (2,4-dichlorophenoxy)acetate (DDA)(2,4-D), didecyldimethylammonium (4-chloro-2-methylphenoxy)acetate (DDA)(MCPA), as well as reference herbicides (4-chloro-2-methylphenoxy)acetic acid (MCPA) and (2,4-dichlorophenoxy)acetic acid (2,4-D) in the form of sodium salts on biogas production efficiency was investigated. The effective concentration (EC50) values were determined for all tested compounds. (MCPA)^−^ was the most toxic, with an EC50 value of 38.6–41.2 mg/L. The EC50 for 2,4-D was 55.7–59.8 mg/L. The addition of the test substances resulted in changes of the population structure of the microbiota which formed the fermentation pulp. The research was based on 16S rDNA analysis with the use of the Next Generation Sequencing method and the MiSeq platform (Illumina, San Diego, CA, USA). There was a significant decrease in bacteria belonging to *Firmicutes* and Archaea belonging to *Euryarchaeota*. A significant decrease of the biodiversity of the methane fermentation microbiota was also established, which was expressed by the decrease of the operational taxonomic units (OTUs) and the value of Shannon’s entropy. In order to determine the functional potential of bacterial metapopulations based on the 16SrDNAprofile, the PICRUSt(Phylogenetic Investigation of Communities by Reconstruction of Unobserved States)tool was used, which allowed to determine the gene potency of microorganisms and their ability to biodegrade the herbicides. In the framework of the conducted analysis, no key genes related to the biodegradation of MCPA or 2,4-D were found, and the observed decrease of their content in the supernatant liquid was caused by their sorption on bacterial biomass.

## 1. Introduction

Anaerobic digestion is a process that uses anaerobic microorganisms, which is widely utilized for processing vegetable biomass into biogas [1]. Methane produced by methanogenic microorganisms belonging to Archaea is the most important component of biogas. The efficiency of biogas production depends on providing optimal conditions for the functioning of the entire microbial consortium involved in the bioconversion of plant biomass into biogas. The process is realized in four stages: Hydrolysis, acidification, acetogenesis and methanogenesis, which are carried out by various groups of Bacteria and Archaea [2].

The correct balance of carbon, nitrogen, phosphorus and sulphur sources, as well as micronutrients such as iron, nickel, molybdenum, tungsten, selenium or cobalt, are extremely important for the proper development and functioning of microorganisms involved in anaerobic digestion [3,4,5]. In addition to a decrease of process efficiency resulting from an excess or deficiency of nutrients, the causes of irregularities in biogas production may include inhibitors, namely substances which decrease the rate or completely stop the biochemical processes carried out by microorganisms [6]. With an excess of nitrogen introduced into the fermentation chamber, harmful ammonia is formed, which reduces the methane production rate, even at a small concentration [7,8]. Other inhibitors include hydrogen sulphide, heavy metals and volatile fatty acids [9]. Additionally, substances widely used in animal breeding, including large-scale veterinary medicines, particularly antibiotics, also exhibit an adverse effect [10].

Herbicides are another important group of compounds used in high quantities in agricultural production. However, due to their numerous undesirable environmental effects, particularly their possible accumulation, there is an ongoing search for substances with herbicidal properties characterized by low toxicity towards a wide range of organisms and new methods of their application [11]. The novel idea of applying active substances in the form of herbicidal ionic liquids (HILs) is particularly attractive due to potential reductions of herbicide use [12], possibility to incorporate compounds of natural origin [13] and introduction of dual pesticidal functions [14].

Ionic liquids belong to a relatively new class of compounds which occur as liquids at the temperature below 100 °C. They consist of a cation and an anion, which determine the traits and properties of a given ionic liquid. These substances are characterized by very low vapor pressure and unique properties due to interactions between ions based on Coulomb and Van der Waals forces, as well as hydrogen bonds. An important feature of these compounds is their ability to dissolve both organic and inorganic materials. Due to this reason they are applied in the chemical, electrochemical, pharmaceutical and biotechnological industries, mainly in the extraction and separation processes of various compounds [15]. Herbicidal ionic liquids (HILs) are a relatively new type of ionic liquids. These compounds possess an ionic structure and consist of a plant growth inhibitor or herbicide. They exhibit biological activity and can be used in lower doses while maintaining high efficiency. Proper selection of the cation in the herbicidal ionic liquid eliminates the necessity to use auxiliary substances, e.g., adjuvants, which are an additional hazard for the natural environment and increase the costs of agricultural production [16,17]. HILs, included in the third generation of ionic liquids, are soluble in water and do not require the addition of organic solvents, which reduces their potential toxicity [18,19]. The research regarding the effectiveness of ionic liquids comprising (4-chloro-2-methylphenoxy)acetic acid (MCPA) in the form of an anion and pyridinium or ammonium cations confirmed higher herbicidal effectiveness of such compounds in comparison to standard, commercially available MCPA salts [20]. Moreover, in the case of HILs comprising (2,4-dichlorophenoxy)acetic acid (2,4-D) in the form of an anion, increased efficiency was observed compared to the available commercial preparations. The tested HILs were chemically and thermally stable. They also exhibited lower solubility in water, which reduces their mobility in soil [17]. These innovative plant protection products may set new trends in crop protection in the future.

However, in addition to their anti-weed effectiveness, HILs are often characterized by high toxicity and may be introduced into groundwater when used at high doses, which creates a potential hazard to humans. The risk of accumulation of herbicides in various elements of the environment may adversely impact the bioconversion processes, in cases where plant biomass which has had contact with HILs is used as the raw material.

The aim of this study was to evaluate the influence of herbicidal ionic liquids on the efficiency of methane production. The potential of anaerobic microorganisms which biodegrade the herbicides, the genetic potential of the environment and changes in microbial communities which form the fermentation pulp were established by determining the indicators of α- and β-biodiversity.

## 2. Materials and Methods

### 2.1. Ionic Liquids

The following ionic liquids were used for the tests: benzalkonium (2,4-dichlorophenoxy)acetate (BA)(2,4-D), benzalkonium (4-chloro-2-methylphenoxy)acetate (BA)(MCPA), didecyldimethylammonium (2,4-dichlorophenoxy)acetate (DDA)(2,4-D), didecyldimethylammonium (4-chloro-2-methylphenoxy)acetate (DDA)(MCPA).The compounds were obtained by chemical synthesis at the Faculty of Technology Chemical University of Technology (Poznan, Poland). Commercially available herbicides in the form of acides were obtained from Sigma-Aldrich (Munich, Germany). Structural formulas of the studied herbicides and herbicidal ionic liquids are presented in Table 1.

### 2.2. Anaerobic Digestion

The fermentation was carried out in glass bioreactors with a capacity of 2 L which were inoculated with 1000 g of inoculum originating from an active agricultural biogas plant. Then, an appropriate amount of HIL ranging from 0 to 125 mg/L was introduced into each bioreactor, respectively, based on the herbicidal component. In the next stage, a carbon source in the form of whey permeate was introduced in the amount of 100 g/L. The initial dry matter content was equal to 76 g/L. After sealing, the bioreactors were placed in a water bath with a constant temperature of 37 °C and mixed several times a day in order to obtain a uniform fermentation pulp. In the upper part of the bioreactor the biogas was discharged by a hose into a 4.5 Lpolycarbonate tube with a scale to measure the amount of gas produced. The analysis of the chemical composition of the gas was carried out using a GA5000 gas analyser from Geotech (Bydgoszcz, Poland) equipped with a head for CO_2_, O_2_, CH_4_, NH_3_ and H_2_S measurements. The analyser was connected to the valve located on the storage tube with a hose. The volume of produced biogas was measured every 24 h and biogas analysis was carried out with a minimum gas volume of 500 cm^3^. All fermentation experiments were carried out in triplicate.

### 2.3. Determination of Herbicidal Ionic Liquids

#### 2.3.1. Sample Preparation

The samples were filtered through a pre-weighed 3h grade filter paper from Munktell&Filtrak (Bärenstein, Germany). The obtained filtrate was diluted with a methanol: water mixture (80:20 v/v) and further filtered through a 0.2 µm polytetrafluoroethylene (PTFE) syringe filter (Agilent Technologies Econofilter from Perlan Technologies, (Santa Clara, USA) before the high-performance liquid chromatography-mass spectrometry (HPLC-MS) analysis, whereas the biomass sludge was dried on the filter paper and weighed. The amount of biomass was calculated by subtracting the filter mass. Next, the filter paper containing the biomass was cut and HILs were extracted using the ultrasound bath with 25 mL methanol for 15 min. The obtained extract was diluted with a methanol: water mixture (80:20 v/v) and filtered through a 0.2 µm PTFE syringe filter before the HPLC-MS analysis.

#### 2.3.2. HPLC-MS Analysis

The HPLC-MS analyses were performed using the UltiMate 3000 RSLC chromatograph from Dionex (Sunnyvale, CA, USA). Two µL samples were injected into a Hypersil GOLD column (100 mm × 2.1 mm L.D. (Length and Diameter); 1.9 µm) with a 2.1 mm L.D. pre-filter cartridge (0.2 µm) from Thermo Scientific (Waltham, MA, USA). The mobile phase consisted of 5 × 10^−3^ mol/L of ammonium formate in water (phase A) and methanol (phase B) at a flow rate of 0.2 mL/ min. Gradient elution was performed by linearly increasing the percentage of phase B from 85 to 100% during 2 min and maintained at 100% for 3 min. The LC column effluent was directed to the API 4000 QTRAP triple quadrupole mass spectrometer from AB Sciex (Foster City, CA, USA) through the electrospray ionization source (Turbo Ion Spray) which operated in positive ion mode for analyses of cations and in negative ion mode for analyses of anions. The dwell time for each mass transition detected in the MS/MS multiple reaction monitoring mode was set to 100 ms. Nitrogen was used as the curtain gas (10 psi), nebulizer gas (40 psi), auxiliary gas (40 psi) and collision gas (medium). The source temperature was equal to 400 °C and the ion spray voltage was at 4500 V for cations and −4500 V for anions. The declustering potential was equal to 50 V for cations and −50 V for anions. The detected mass transitions and specific parameters of each analyte are summarized in Table 2.

### 2.4. Metagenomic Analysis

#### 2.4.1. DNA Extraction

In order to conduct a metabiome analysis, samples of 5 mL of each replicate were taken, mixed together, averaged and used as a starting material for DNA isolation.

DNA extraction: Total DNA was extracted from 1 mL of sample using a Genomic Mini AX Soil kit (A&A Biotechnology, Gdańsk, Polska) according to manufacturer’s instruction. The extracted DNA was quantified using Quant-iT HS ds.-DNA assay kit (Life Technologies, Carlsbad, CA, USA) on Qubit 3 fluorometer; 2 µL of extracts were examined on a 0.8% agarose gel (Life Technologies, Carlsbad, CA, USA).

#### 2.4.2. PCR Amplification

Region IV of bacterial 16S rDNA gene was amplified using universal primers 515F and 806R: containing reverse complement of 3’ Illumina adapter, golay barcode, reverse primer pad, reverse primer linker and reverse primer (Table 1). Genomic DNA (100 ng) was used for PCR amplification in a 50 µL reaction volume containing: 1× PCR reaction mix, 0.25 µM each primer and 5U Taq DNA Polymerase (A&A Biotechnology, Poland). The following conditions were used for PCR amplification: initial denaturation 95 °C for 3 min; 25 cycles of denaturation 30 s at 94 °C, annealing 30 s at 52 °C and extension 2 min at 72 °C and a final extension at 72 °C for 10 min. The products were purified in Clean-Up columns (A&A Biotechnology) according to manufacturer’s protocol. The libraries were constructed from amplicons using NEBNext^®^ DNA Library Prep Master Mix Set for Illumina (New England Biolabs, Hitchin, UK). Then the libraries were pooled at equimolar concentration. Pooled libraries were quantified using a Qubit Fluorimeter and dsDNA HS assay kit (Life Technologies, Carlsbad, CA, USA). The library was denatured with 0.2 N NaOH and diluted with HT1 buffer (Illumina, San Diego, CA, USA) to a final concentration of 8 pM. To balance the overall lack of sequence diversity, a spike-in of denatured Phix was added to the concentration of 40%. Sequencing was conducted on an Illumina MiSeq (Illumina, San Diego, CA, USA) using paired-end (2 × 250) MiSeq Reagent Kits v2 (Illumina, San Diego, CA, USA). Sequencing primers were based on Caporasoet al. [21] (Table 1). The sequencing reaction was performed with MiSeq Illumina instrument and MiSeq Reagent Kit v2 (2 × 250 bp).

#### 2.4.3. Bioinformatic Analysis

The sequencing data was processed using CLC Genomic Workbench 8.5 and CLC Microbial Genomics Module 1.2. (Qiagen Bioinformatics, Aarhus, Denmark). Total number of reads ranged from 182,354 to 223,487. After sequencing, the reads were demultiplexed to the probes, the overlapping paired-end reads were merged (68–76% of total reads) and trimmed to yield fragments of 289 nt. Only fragments which passed the merging were retained for downstream processing. Chimeric reads (from 24,258 to 26,982) were filtered and remaining sequences were assigned to operational taxonomic units (OTUs).

The number of reads which passed merging and trimming ranged from 84,128 to 1,116,529. Reads were clustered against the SILVA v119 99% 16S rDNA gene database [22,23,24]. Rarefaction analysis with a depth of 60,000 sequences per sample was used to calculate Alpha diversity measured using OTU abundance and Shannon’s entropy and Beta diversity (Bray-Curtis principal coordinate analysis-PCoA) parameters were determined

To determine the predicted gene content from each OTU table constructed against GreenGenes 13.5, the PICRUSt (Phylogenetic Investigation of Communities by Reconstruction of Unobserved States) tool was used [25]. The output data was a set of abundance of functional orthologs (KO) in each sample. On the basis of the Kyoto Encyclopedia of Genes and Genomes (KEGG), key orthologs for the biodegradation of MCPA and 2,4-D were selected which allowed the prediction of functional pathways [26]. Finally, the data was grouped against the EC classification (Enzyme Commission number). In order to compare the samples, the results were normalized by dividing the total predicted gene abundance for the number of OTUs in each sample.

## 3. Results and Discussion

### 3.1. The Effect of Herbicidal Ionic Liquids on the Amount of Obtained Biogas

During the anaerobic digestion in the system without the addition of herbicidal ionic liquids (HILs), the process yield reached 260 mL of methane/kg dry mass of sediment. Addition of HILs in an amount up to 125 mg/L(calculated based on the active herbicidal anion) significantly reduced the methane production of the system (Figure 1).

The comparison of the effect of HILs comprising (MCPA)^−^ or (2,4-D)^−^ anions indicated that ionic liquids with the (MCPA)^−^ were more toxic in low concentration, than those with (2,4-D)^−^. Similarly, an inhibition of biogas production occurred when MCPA and 2,4-D were used. In this case, the MCPA herbicide was also more toxic than 2,4-D. The addition of 125 mg/L of the herbicide (calculated based on the active substance) completely inhibited biogas production. The EC50 values determined for the anaerobic digestion process are shown in Table 3.

The obtained data confirm the results of studies conducted by Ławniczak et al. [27], which also reported the higher toxicity of (MCPA)^−^ compared to (2,4-D)^−^ towards aerobic bacteria from different environments. However, the determined EC50 values for MCPA and 2,4-D were higher than those obtained in this work. It can therefore be concluded that anaerobic microorganisms are more sensitive to the presence of herbicides compared to aerobic bacteria. Similar results were obtained by Sanchis et al. [28] in the framework of research regarding MCPA and 2,4-D toxicity towards activated sludge. The EC50 values for the tested herbicides were equal to 144 and 213 mg/L, respectively, and were also higher than for anaerobic microorganisms. However, it should be noted that the toxicity of both MCPA and 2,4-D towards plant and animal cells is different than for bacteria. In this case, the toxicity of 2,4-D is higher than that of MCPA [29].

### 3.2. Biodegradation of Herbicidal Ionic Liquids

During the course of anaerobic digestion, the residues of herbicides were determined and their concentration was analysed both in the sludge and in the supernatant. The initial concentration of HILs added to the fermentation pulp was 50.0 mg/L (calculated based on the active substance 2,4-D and MCPA). In the case of herbicides, their presence was found both in the liquid and in the sediment. However, their concentration did not differ statistically. A higher concentration of herbicides was observed in the supernatant. Their amount absorbed on the sediment biomass was much lower and did not exceed 10 mg/kg. The determined concentrations of herbicides in the fermentation pulp lead to the conclusion that these compounds were not biodegraded and the decrease of their concentration in the supernatant was related to their adsorption on bacterial biomass (Table 3). Other authors also observed the presence of herbicides and other biologically active substances used in plant protection in the post-fermentation pulp [30,31].

### 3.3. Metapopulation Analysis

Taxonomic identification based on the hypervariable region of 16s rDNA using the SILVA v119 database allowed for the detection of microorganisms comprising the fermentation pulp. In all tests, both Bacteria and Archaea were found, which belonged to 82 classes (Figure 2).

In the fermentation pulp without the addition of herbicides (control), the ratio of Archaea was highest and amounted to 32%. *Methanobacteria* was the dominant class among Archaea (25% of Archaea). *Clostridia* was the predominant Bacteria domain-related class (37% of Bacteria). Such composition is characteristic for microbial communities which carry out the anaerobic digestion process [32,33]. The addition of both HILs and commercial herbicides caused changes in the population structure of Bacteria and Archaea. *Firmicutes* and *Euryarchaeota* were particularly sensitive phyla to the presence of herbicides. The decrease in the ratio of Archaea belonging to the *Methanobacteria* class in all variants of the experiment in comparison to the control sample was particularly noticeable. Their content decreased to 3% after addition of (DDA)(MCPA).

A particularly toxic effect was observed after the addition of MCPA in case of the *Methanomicrobia* class, where the ratio was decreased to 0.45–2.3%.

This was caused by the extremely high sensitivity of Archaea belonging to the *Methanosaeta* genus (which was predominant in this class of microorganisms) to the presence of herbicides. The addition of MCPA and HILs with (MCPA)^−^ to the fermentation pulp in an amount of 50 mg/L resulted in its complete elimination from the community of microorganisms conducting the methanogenesis process. In contrast, the addition of 2,4-D or HILs with (2,4-D)^−^ caused a 52–76% decrease in its ratio with respect to the anaerobic digestion process without the addition of herbicides.

The introduction of the herbicide in the form of HILs containing (MCPA)^−^ and (2,4-D)^−^ also increased the toxicity and caused a significant decrease of both *Methanobacteria* and *Clostridia* classes. The *Methanobacterium* genus (which was predominant in the *Methanobacteria* class) was characterized by higher resistance to the presence of herbicides compared to the *Methanosarcina* genus (which was predominant in the *Methanomicrobia* class). In this case, a decrease of its ratio in the population was observed and the presence of herbicides did not eliminate it from the environment. Based on the comparison of the taxonomic composition of the pulp, the most notable changes caused by the addition of HILs concerned bacteria belonging to the *Bacilli* and *Clostridia* classes. However, in all cases, the increase of the percentage ratio of the Proteobacteria class in relation to the metapopulation of the fermentation pulp can be noticed. The decrease of the ratio of *Bacilli* and *Clostridia* classes in the microbial community caused an increase of the ratio of *Bacteroidetes* phylum (the abundance of which in the control sample amounted to 0.3%). These bacteria were characterized by very high resistance to the presence of herbicides and HILs. As a result, their ratio in the population increased after addition of 2,4-D and MCPA to 6.5% and 9.1%, respectively. On the other hand, the addition of herbicides in the form of HILs resulted in the increase of their abundance in the microbial community from 32.0% to 41%. The *Porphyromonadaceae* and *Prevotellaceae* families as well as numerous uncultured bacteria showed particularly high resistance to herbicides. The fermentation pulp is a dynamic multi-phase system, which contributes to the diversity of the local metapopulation structure of indigenous microorganisms, and thus their metabolic profiles. Alpha-diversity analysis expressed as number of OTUs showed significant differences between all variants of experiments (Table 4) compared to the control experiment.

This indicates a significant decrease of biodiversity in the fermentation pulp caused by the addition of herbicidal substances. The addition of HILs with (MCPA)^−^ resulted in a greater reduction of biodiversity than the addition of HILs with (2,4-D)^−^. This phenomenon is often described in the literature and is related to the influence of toxic substances which change the composition of the microflora. This is particularly true for environments characterized by high anthropopressure, especially in heavily polluted areas, where a smaller number of OTUs is found than in pollution-free environments. This may be due to the fact that only microorganisms with enzymatic profiles specialized for biodegradation of xenobiotics and characterized by high resistance to the existing stress factors are able to grow as a result of long-term exposure to pollutants. Often, the decline in biodiversity may also be related to the formation of dead-end products during the biodegradation of various xenobiotics, which have a toxic effect on some of the metapopulations in the environment [34].

The Principal coordinates analysis (PCoA) with Bray-Curtis dissimilarity (Figure 3) also showed significant differences in case of populations which carried out the fermentation process in the presence of herbicides, compared to the control sample.

The most notable changes in comparison to the composition of the control population occurred after the addition of (BA)(MCPA) and (BA)(2,4-D) into the pulp. The distances between the samples presented in Figure 3 were the highest, hence it can be concluded that the anion originating from HILs is responsible for changes in the population structure of methanogenic microorganisms. This is also confirmed in case of the pulp with the addition of (BA)(MCPA) and (BA)(2,4-D). However, these distances are lower, suggesting lower differences in the microbial population structure in these experimental variants. Additionally, the changes in meta-populations of microorganisms caused a specific division into two groups of microorganisms separated from each other depending on the type of herbicide added.

Evaluation of distances in the Bray-Curtis dissimilarity analysis for the addition of herbicidal ionic liquids indicated that the changes in the population of microorganisms resulting from the influence of HILs were targeted and non-random.

#### Predicted functional gene abundance

Metapopulations of microorganisms are characterized by the dynamics of qualitative and quantitative changes under the influence of varying environmental parameters and exposure to toxic compounds [35]. The PICRUSt tool was usedin order to predict the functional potential of bacterial metapopulations based on the 16S DNA profile. Reference genome coverage for all samples was calculated using weighted NSTI (Nearest Sequenced Taxon Index) score. The NSTI for all trials was in the range of 0.04–0.09, which proves a good availability of the genomes reference closely related to the microorganisms in the sample [25]. The total predicted gene abundance divided for the number of OTUs in each sample was presented in Figure 4 and in Table 5.

Prediction of the presence of the sum of genes responsible for the biological degradation of 2,4-D and MCPA indicated a decrease in their number relative to the control (Table 5). In all the tests, a relative zero abundance of enzymes leading the first biodegradation reaction of herbicides was found: tfdA (EC 1.14.11.-) and tfdB (EC 1.14.13.20).

This was confirmed by the biodegradation results, in which no decrease of herbicide concentration was established during the anaerobic digestion process. In order to explain this phenomenon, the PICRUSt program algorithm should be taken into consideration. During categorization to the metabolic pathway level, all genes included in it are counted (based on the KEGG database). However, taking into account the selected orthologs units, a clear tendency of decline in the abundance of the majority of crucial orthologs in the first stages of biodegradation of herbicides (e.g., the genes encoding dichlorophenol monooxigenase: tfdA and tfdB) can be observed. It is considered that the moment of ring splitting is the key stage of biodegradation of numerous xenobiotics, which determines the efficiency of the process. In associated with the above, it can be assumed that the lack of or low number of one of the first enzymes may result in the inhibition of the biodegradation processes, despite the generally high indications at the level of the metabolic pathway.

The presence of some genes responsible for the biodegradation of herbicides is related both to the fact of coexistence of various metabolic pathways of the enzymatic degradation of xenobiotics as well as the limitations associated with insufficient data in the databases which are the basis for the operation of bioinformatic algorithms. In order to use the estimation method based on 16SrDNA data, it is worth conducting an analysis of all available orthologs. This also refers to the construction of other molecular tools for the genetic evaluation of biodegradation potential, e.g., microarrays. Comprehensive multivariate analysis may contribute to the increased effectiveness of such tools. The dynamics of microbial metapopulation changes under the influence of herbicide contamination is undoubtedly a targeted process, however, due to the complexity of the microbial community, more research is needed in this field.

Biodegradation of HILs is possible under aerobic conditions both in soil [20] and in the aquatic environment [36]. Sydow et al. observed changes in the composition of the microbiota resulting in the increase of *Proteobacteria*, *Actinobacteria* and *Bacteroidetes* in the microbial community with a simultaneous decrease of the *Firmicutes* ratio. Additionally, it was reported that phosphonic-based HILs can be a stress factor for soil microorganisms and affect their biodiversity changes, particularly by increasing the ratio of microorganisms well known for biodegradation of hydrocarbons, such as the *Sphingomonas* and *Pseudomonas* genera [36]. Other authors conducted the biodegradation of herbicides in the aquatic environment with the use of activated sludge. However, this process was preceded by pre-oxidation with the use of ozone, which significantly increased the biodegradation efficiency [37]. On the other hand, the removal of herbicides under anaerobic conditions was only possible with the use of a wastewater treatment system consisting of a combination of aerobic and anaerobic processes using the sequencing batch bioreactor (SBR) [38] and an acclimation period of 70 days. The use of such approach allowed for complete removal of 2,4-D from sewage with a concentration of 500 mg/L.

## 4. Conclusions

The dynamics of metapopulation changes caused by the presence of HILs in the fermentation pulp is a targeted process, which results in a decrease of methane production efficiency mainly due to the reduction of the ratio of Archaea responsible for the production of methane. Due to the lack of key genes related to the biodegradation of MCPA and 2,4-D, the microorganisms which carry out the methane fermentation process were exposed to the phenomenon of accumulation of herbicides in the biomass. This causes changes in the microbial population structure and, consequently, inhibits methane production. However, due to the complexity of the fermentation environment, further research into the toxicity of herbicides in relation to the anaerobic microbiota is necessary.

## Figures and Tables

**Figure 1 ijerph-16-00916-f001:**
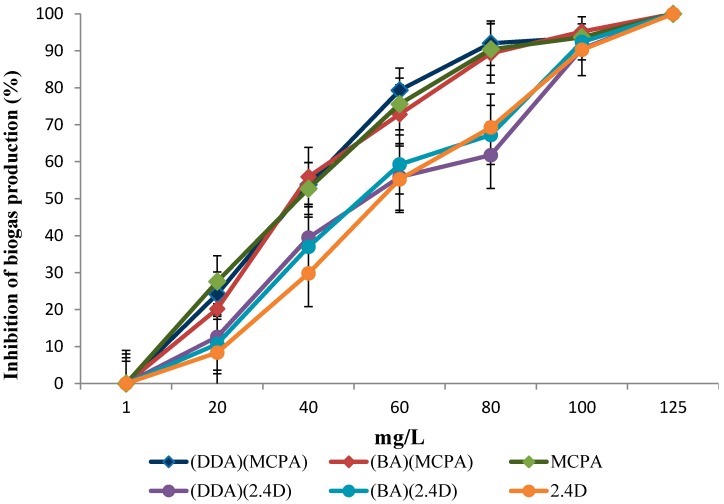
The effect of selected ionic liquids (benzalkonium (2,4-dichlorophenoxy)acetate (BA)(2,4-D), benzalkonium (4-chloro-2-methylphenoxy)acetate (BA)(MCPA), didecyldimethylammonium (2,4-dichlorophenoxy)acetate (DDA)(2,4-D), didecyldimethylammonium (4-chloro-2-methylphenoxy)acetate (DDA)(MCPA) as well as 4-chloro-2-methylphenoxy)acetic acid (MCPA) and 2,4-dichlorophenoxy)acetic acid (2,4-D) on the inhibition of biogas production. All fermentation experiments were carried out in triplicate.

**Figure 2 ijerph-16-00916-f002:**
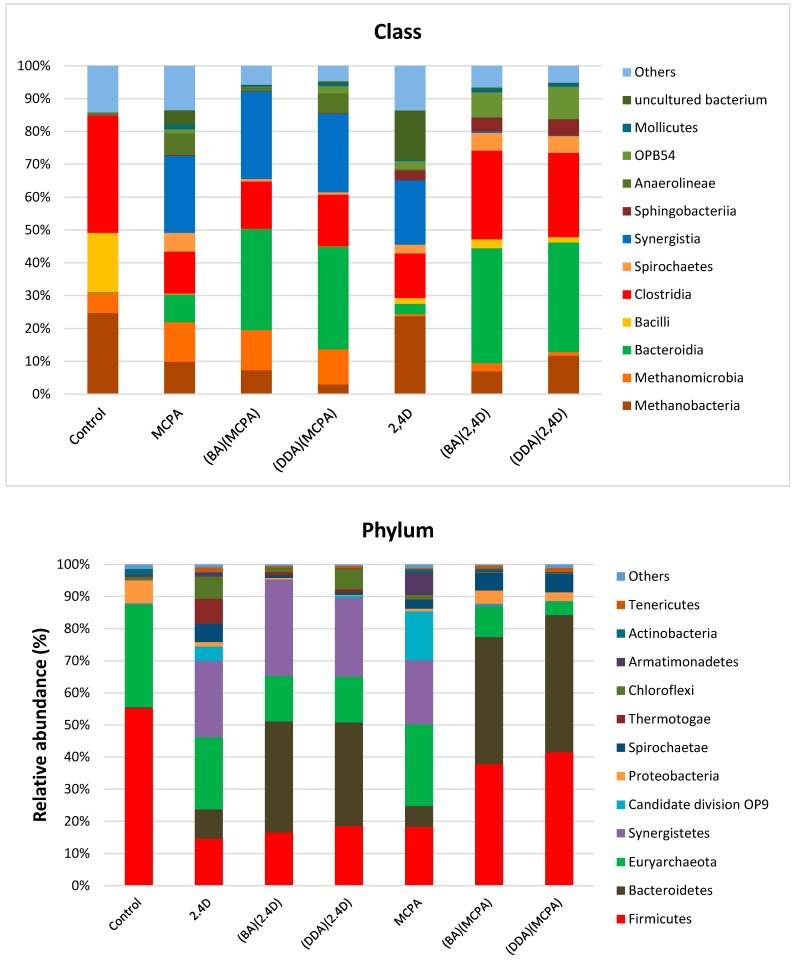
The ratio of Classes and Phyla in the biogas-producing microbial communities.

**Figure 3 ijerph-16-00916-f003:**
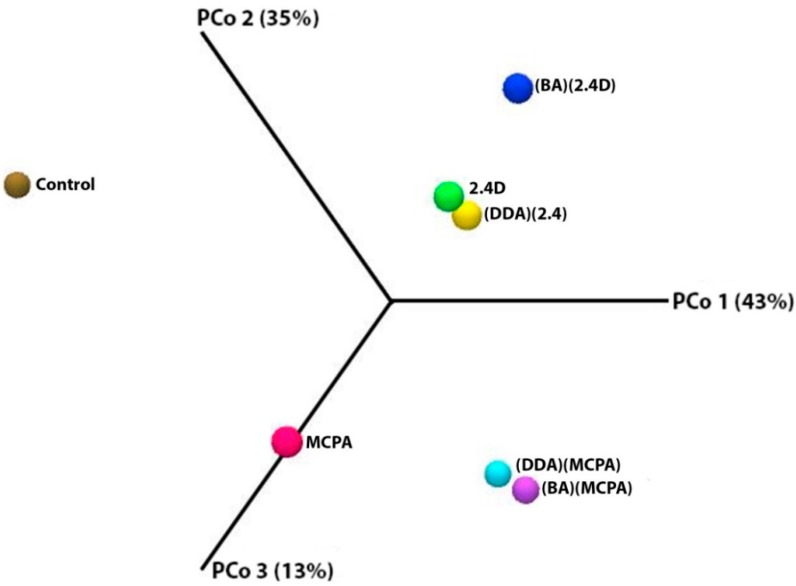
Coordinates analysis (PCoA) based on Bray-Curtis index for biogas-producing microbial community in the presence of HILs.

**Figure 4 ijerph-16-00916-f004:**
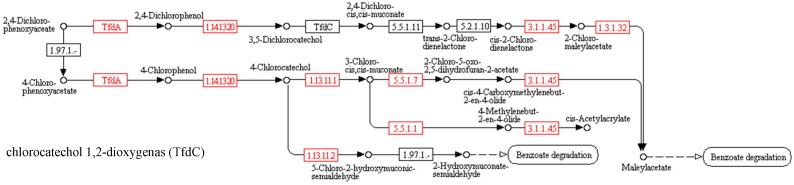
List of orthologs which participated in the MCPA and 2,4-D biodegradation process included in the KEGG (Kyoto Encyclopedia of Genes and Genomes) database.

**Table 1 ijerph-16-00916-t001:** Characteristics of tested herbicidal ionic liquids (HILs) and herbicides.

HIL/Herbicide	Chemical Structure	Molecular Mass (g/mol)
(BA)(2,4-D)	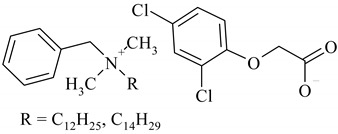	540.64
(BA)(MCPA)	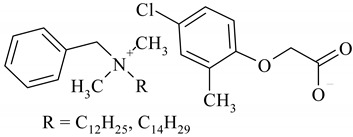	520.27
(DDA)(2,4-D)	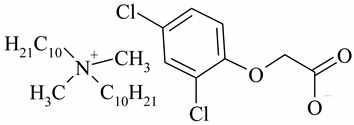	546.04
(DDA)(MCPA)	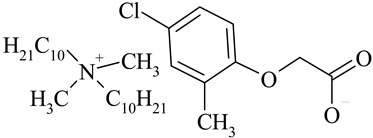	526.33
2,4-D	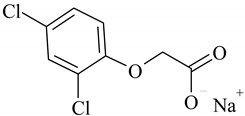	243.02
MCPA	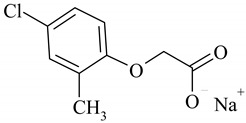	222.61

**Table 2 ijerph-16-00916-t002:** Parameters of mass spectrometric detection characteristic for the particular analytes. Analytical multiple reaction monitoring transition (MRM 1). Confirmatory multiple reaction monitoring transition (MRM 2).

Analyte	MRM Transitions (Precursor Ionm/z→Product Ion m/z)			
	MRM 1	Collision Energy (V)	MRM 2	Collision Energy (V)
(BA)	304→91	35	304→212	29
(DDA)	326→186	38	-	-
(MCPA)	199→141	−23	199→155	−15
(2,4-D)	219→161	−30	219→125	−37

**Table 3 ijerph-16-00916-t003:** EC50 values and concentration (mg/L) of HILs in the supernatant and biomass given as mean values, standard deviations of the mean.

Herbicidal Ionic Liquids and Herbicides	EC50	Liquid		Biomass	
		Cation	Anion	Cation	Anion
(DDA)(MCPA)	38.6	4.1 ± 1.4	44.3 ± 6.9	52.5 ± 6.5	2.8 ± 1.6
(DDA)(2,4-D)	59.8	2.4 ± 1.6	37.4 ± 5.2	40.6 ± 4.6	6.4 ± 3.6
(BA)(2,4-D)	55.7	3.3 ± 2.3	45.4 ± 8.7	5.2 ± 2.1	1.7 ± 0.9
(BA)(MCPA)	40.2	0.4 ± 0.3	45.0 ± 9.3	42,0 ± 2.1	7.3 ± 2.8
MCPA	45.7	-	38.6 ± 6.8	-	9.3 ± 5.3
2,4-D	65.8	-	48.6 ± 7.9	-	6.2 ± 4.6

*N* (number of replicates) =3.

**Table 4 ijerph-16-00916-t004:** Values of alpha-biodiversity ratios of microbial metapopulation after 30 days of fermentation.

Alpha-Biodiversity Ratios	Control	MCPA	(BA)(MCPA)	(DDA)(MCPA)	2,4-D	(BA)(2,4-D)	(DDA)(2,4-D)
Number of OTU	366	291	295	275	296	301	317
Shannon’s entropy	5.4	4.4	4.45	4.6	4.9	4.8	5.0

**Table 5 ijerph-16-00916-t005:** Relative abundance of predicted metagenome (%).

EC Number	Abbreviations of Names of Enzymes	Names of Enzymes	Control	MCPA	(BA)(MCPA)	(DDA)(MCPA)	2,4D	(BA)(2,4-D)	(DDA)(2,4-D)
EC 1.14.11.-	tfdA	alpha-ketoglutarate-dependent 2,4-dichlorophenoxyacetate dioxygenase	0.30	0.00	0.00	0.00	0.01	0.12	0.07
EC 1.14.13.20	tfdB	2,4-dichlorophenol 6-monooxygenase	0.01	0.00	0.00	0.00	0.00	0.02	0.00
EC 3.1.1.45	E3.1.1.45 (tfdE)	carboxymethylenebutenolidase	8.03	10.98	5.48	7.32	2.35	3.62	2.91
EC 5.5.1.1	catB	muconatecycloisomerase	2.65	0.52	0.29	0.19	0.60	1.06	1.18
EC 5.5.1.7	E5.5.1.7 (tfdD)	chloromuconatecycloisomerase	0.00	0.00	0.00	0.00	0.00	0.00	0.00
EC 1.3.1.32	E1.3.1.32 (tfdF)	maleylacetate reductase	0.33	0.01	0.00	0.00	0.03	0.15	0.07
EC 1.13.11.1	catA	catechol 1,2-dioxygenase	0.54	0.03	0.00	0.01	0.07	0.51	0.31
EC 1.13.11.2	dmpB	catechol 2,3-dioxygenase	0.52	3.47	0.47	3.21	0.30	0.42	0.27
EC 1.13.11.2	catE	catechol 2,3-dioxygenase	1.69	3.64	0.64	3.22	0.87	0.73	0.30

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
