# Peer review of "The Toxic Effect of Herbicidal Ionic Liquids on Biogas-Producing Microbial Community"

_ijerph, 2019, doi:10.3390/ijerph16060916_

Round 1

Reviewer 1 Report

In the present manuscript, the authors describe the effect of some herbicidal ionic liquids (HILs) on the composition of microbial communities taking part in biogas production.  Results are interesting and the study provides new knowledge on the ecotoxicological effects of the investigated herbicides. However, several parts of the manuscript should be significantly changed. Analysis of microbiological data is not sufficient, and both scientific terminology and English should be improved.

Major comments:

1.      The authors write that every fermentation was carried out in triplicates. In this case, it is not clear which of the parallel experiments were chosen for 16S rDNA amplicon sequencing and how the microbial communities of the parallel experiments were related to each other.

2.      The part of the manuscript that deals with the description of microbial communities and discuss the related results is unsatisfying. I see many more differences between the communities, which should be also mentioned (e.g. the control sample almost lacks Bacteroidetes, while in some of the treated communities these bacteria are highly abundant). Moreover, I do not see why the authors say that the increased ratio of Proteobacteria can be noticed in the treated samples. This increase is negligible compared to other changes. This section of the manuscript should be rewritten.

3.      The authors performed some statistical analyses e.g. PCoA, but these methods are not mentioned in the M&M section of the manuscript.

4.      The English grammar and the microbiological terminology of the text should be significantly improved. There are several lengthy and diffuse sentences in the text that should be simplified in my opinion. I tried to fix some of the terminological mistakes which are mentioned below, but I think that there are many more in the manuscript.

Minor coments:

1.      Line 25: Abstract – Do not use the term microflora. Replace it with microbiota.

2.      Lines 25-26: “The research was based on 16S RNA analysis…” – I think the authors analysed 16S rDNA amplicons and not 16S rRNA.

3.      Line 27: Fermicutes?

4.      Line 31: 16S rRNA gene or 16S rDNA. Please use the proper terminology throughout the manuscript!

5.      Line 40: Change methane microorganisms to methanogenic microorganisms.

6.      Lines 80-81: increased efficiency was observed…

7.      Line 147 – assay kit

8.      Line 155 – To get amplicons for the Illumina sequencing the authors applied considerably high cycle numbers (35) during the PCR. What was the reason of such high cycle number, knowing that the optimal cycle number is 25?

9.      Line 160 – assay kit

10.   Lines 190-192: This sentence is way too complicated! Please try to simplify it. A good example would be the following: Addition of HILs in an amount of up to 125 mg/L (calculated based on the active herbicidal ion) significantly reduced the methane production of the system.

11.   Line 199: 125 mg/L?

12.   Table 3. Biomasa?

13.   Lines 232-233: The ratio of Archaea…

14.   Line 233: …class belonging to Archaea…

15.   Lines 234-235: The sentence here should be changed as follows: “The most dominant Bacteria domain-related class was the Clostridia (37% of Bacteria or x % of total sequence reads). I suggest highlighting the % of total sequence reads to be comparable with archaebacterial sequence read ratios.

16.   Line 237: …structure of Bacteria and Archaea.

17.   Fig. 2. …microbial communities.

18.   Lines 250-253: These sentences are a bit chaotic for me.

Author Response

In the present manuscript, the authors describe the effect of some herbicidal ionic liquids (HILs) on the composition of microbial communities taking part in biogas production.  Results are interesting and the study provides new knowledge on the ecotoxicological effects of the investigated herbicides. However, several parts of the manuscript should be significantly changed. Analysis of microbiological data is not sufficient, and both scientific terminology and English should be improved.

 Answer: We agree with the reviewer’s remark. The manuscript has been corrected.

Major comments:

1.      The authors write that every fermentation was carried out in triplicates. In this case, it is not clear which of the parallel experiments were chosen for 16S rDNA amplicon sequencing and how the microbial communities of the parallel experiments were related to each other.

Answer. IWe agree with the reviewer’s remark. In order to conduct a metabiome analysis, samples of 5 ml of each replicate were taken, mixed together, averaged and used as a starting material for DNA isolation.

The manuscript has been corrected.  Page 5 Line 159-161.

2.      The part of the manuscript that deals with the description of microbial communities and discuss the related results is unsatisfying. I see many more differences between the communities, which should be also mentioned (e.g. the control sample almost lacks Bacteroidetes, while in some of the treated communities these bacteria are highly abundant). Moreover, I do not see why the authors say that the increased ratio of Proteobacteria can be noticed in the treated samples. This increase is negligible compared to other changes. This section of the manuscript should be rewritten.

Answer: We agree with the reviewer’s remark. The manuscript has been corrected.

3.      The authors performed some statistical analyses e.g. PCoA, but these methods are not mentioned in the M&M section of the manuscript.

Answer: We agree with the reviewer’s remark. The manuscript has been corrected. Page 6 Line 193-196.

4.      The English grammar and the microbiological terminology of the text should be significantly improved. There are several lengthy and diffuse sentences in the text that should be simplified in my opinion. I tried to fix some of the terminological mistakes which are mentioned below, but I think that there are many more in the manuscript.

Answer: We agree with the reviewer’s remark. The manuscript has been corrected.

Minor coments:

1. Line 25: Abstract – Do not use the term microflora. Replace it with microbiota.

Answer: We agree with the reviewer’s remark. The manuscript has been corrected. Line 24-25.

2.Lines 25-26: “The research was based on 16S RNA analysis…” – I think the authors analysed 16S rDNA amplicons and not 16S rRNA.

Answer: We agree with the reviewer’s remark. The manuscript has been corrected.

3.Line 27: Fermicutes?

Answer: We agree with the reviewer’s remark. The manuscript has been corrected. Line 27.

4.Line 31: 16S rRNA gene or 16S rDNA. Please use the proper terminology throughout the manuscript!

Answer: We agree with the reviewer’s remark. The manuscript has been corrected.

5.Line 40: Change methane microorganisms to methanogenic microorganisms.

Answer: We agree with the reviewer’s remark. The manuscript has been corrected. Line 159-161.

6.Lines 80-81: increased efficiency was observed…

Answer: We agree with the reviewer’s remark. The manuscript has been corrected. Line 84-87.

7.Line 147 – assay kit

Answer: We agree with the reviewer’s remark. The manuscript has been corrected. Line 165.

8.      Line 155 – To get amplicons for the Illumina sequencing the authors applied considerably high cycle numbers (35) during the PCR. What was the reason of such high cycle number, knowing that the optimal cycle number is 25?

Answer: We agree with the reviewer’s remark. The manuscript has been corrected. Line 173.

9.      Line 160 – assay kit

Answer: We agree with the reviewer’s remark. The manuscript has been corrected. Line 173.

10.   Lines 190-192: This sentence is way too complicated! Please try to simplify it. A good example would be the following: Addition of HILs in an amount of up to 125 mg/L (calculated based on the active herbicidal ion) significantly reduced the methane production of the system.

Answer: We agree with the reviewer’s remark. The manuscript has been corrected. Line 209-212.

11.   Line 199: 125 mg/L?

Answer: We agree with the reviewer’s remark. The manuscript has been corrected. Line 220.

12.   Table 3. Biomasa?

Answer: We agree with the reviewer’s remark. The table 3 has been corrected.

13.   Lines 232-233: The ratio of Archaea…

Answer: We agree with the reviewer’s remark. The manuscript has been corrected. Line 260.

14.   Line 233: …class belonging to Archaea…

Answer: We agree with the reviewer’s remark. The manuscript has been corrected. Line 261.

15.   Lines 234-235: The sentence here should be changed as follows: “The most dominant Bacteria domain-related class was the Clostridia (37% of Bacteria or x % of total sequence reads). I suggest highlighting the % of total sequence reads to be comparable with archaebacterial sequence read ratios.

Answer: We agree with the reviewer’s remark. The manuscript has been corrected. Line 262-265.

16.   Line 237: …structure of Bacteria and Archaea.

Answer: We agree with the reviewer’s remark. The manuscript has been corrected. Line 266-267.

17.   Fig. 2. …microbial communities.

Answer: We agree with the reviewer’s remark. The manuscript has been corrected. Line 275.

18.   Lines 250-253: These sentences are a bit chaotic for me.

Answer: We agree with the reviewer’s remark. The manuscript has been corrected. Line 291-294.

Reviewer 2 Report

Please see attached document for minor editing reviews and comments.

in particular, please review the statement on the control in Section 3.2.

Please also consider the presentation of ratios in Section 3.3

Author Response

Please see attached document for minor editing reviews and comments.

in particular, please review the statement on the control in Section 3.2.

Please also consider the presentation of ratios in Section 3.3

Answer: We agree with the reviewer’s remark. The manuscript has been corrected.

Reviewer 3 Report

Dear Authors, 

The manuscript "The toxic effect of herbicidal ionic liquids on biogas-producing microbial community" is interesting for the scientific community but some aspects must be improved before considered for publication. The writing of the manuscript (genaral wording) should be improved.

Specific comments:

line 104: please, write properly the units (dm3, not dm3), the same in line 111, 113, 116....please check all the manuscript 

Paragraph 2.2: Did you add water? wich amount? describe it better 

Paragraph 2.3.1: Which paper filter? please specify it better and improve this paragraph, it is a bit hard to understand

line 189: now you are using cm3, please use always International Units 

Fig 1: How many replicates did you use? please add this info in the legend. Add error bars in the figure as well 

196-197: this information is not always true. At high concentrations is almost the same. Be always precise

line 198: which commercial herbicides? be precise 

199: 125 mg/?

201: "were shown..."  better: are shown....

Table 3: please describe it better (units, how many replicates, +-? is it SD or SE?...)

line 214: "methane fermentation"? this is not a good term. You get methane through a fermentation process but you do not ferment methane, please check all the manuscript

222-226: Are you sure? How do you know it? you could know it through a Mass Balance but you didn't.

line 234: residues? do you mean the rest of...? 

236: "methane fermentation process" better anaerobic digestion process...

Fig. 2:  improve the colors, it is difficult to distinguish the different colors (blue for example)

Fig. 3 and 4: please discuss deeper your results 

line 305: check your reference (Ławniczak et al.X)

322-323. which genes? did you research this genes to be sure that these are missing? 

Author Response

The manuscript "The toxic effect of herbicidal ionic liquids on biogas-producing microbial community" is interesting for the scientific community but some aspects must be improved before considered for publication. The writing of the manuscript (genaral wording) should be improved.

Answer: We agree with the reviewer’s remark. The manuscript has been corrected.

Specific comments:

line 104: please, write properly the units (dm3, not dm3), the same in line 111, 113, 116....please check all the manuscript 

Answer: We agree with the reviewer’s remark. The manuscript has been corrected.

Paragraph 2.2: Did you add water? wich amount? describe it better 

The initial dry matter content was 76g / L. Water wasn’t added. Line 113-114.

Paragraph 2.3.1: Which paper filter? please specify it better and improve this paragraph, it is a bit hard to understand

Answer: We agree with the reviewer’s remark. The manuscript has been corrected. Line 130-137.

line 189: now you are using cm3, please use always International Units 

Answer: We agree with the reviewer’s remark. The manuscript has been corrected. Line 208.

Fig 1: How many replicates did you use? please add this info in the legend. Add error bars in the figure as well 

Answer: We agree with the reviewer’s remark. The manuscript has been corrected.

196-197: this information is not always true. At high concentrations is almost the same. Be always precise

Answer: We agree with the reviewer’s remark. The manuscript has been corrected.  Line 218.

line 198: which commercial herbicides? be precise 

Answer: Similarly, an inhibition of biogas production occurred when preparations containing 2,4-D or MCPA were used. The manuscript has been corrected. Line 219-220.

199: 125 mg/?

Answer: We agree with the reviewer’s remark. The manuscript has been corrected. Line 220.

201: "were shown..."  better: are shown....

Answer: We agree with the reviewer’s remark. The manuscript has been corrected. Line 222.

Table 3: please describe it better (units, how many replicates, +-? is it SD or SE?...)

Answer: We agree with the reviewer’s remark. The Table 3 has been corrected.

line 214: "methane fermentation"? this is not a good term. You get methane through a fermentation process but you do not ferment methane, please check all the manuscript

Answer: We agree with the reviewer’s remark. The manuscript has been corrected.

222-226: Are you sure? How do you know it? you could know it through a Mass Balance but you didn't.

Answer: We agree with the reviewer’s remark. The manuscript has been corrected. Line 252-253.

line 234: residues? do you mean the rest of...? 

Answer: We agree with the reviewer’s remark. The manuscript has been corrected. Line 261-263.

236: "methane fermentation process" better anaerobic digestion process...

Answer: We agree with the reviewer’s remark. The manuscript has been corrected. Line 265.

Fig. 2:  improve the colors, it is difficult to distinguish the different colors (blue for example)

Answer: We agree with the reviewer’s remark. The Figure 2 has been corrected.

Fig. 3 and 4: please discuss deeper your results 

Answer: We agree with the reviewer’s remark. The manuscript has been corrected.

line 305: check your reference (Ławniczak et al.X)

Answer: We agree with the reviewer’s remark. The manuscript has been corrected. Line 387

322-323. which genes? did you research this genes to be sure that these are missing? 

Answer: We agree with the reviewer’s remark. The manuscript has been corrected. Line 340-400.

Round 2

Reviewer 1 Report

The authors have significantly improved the quality of the manuscript. However, some minor modifications are still needed.

In case of Figure 2 the assignment of the colours is incomplete! Class of Methanobacteria is missing in case of the class level diagram, while at least two phyla are missing in case of the phylum level diagram (red and dark blue phyla).

Line 260: "The Methanomicrobia genus..." Such genus does not exist! Should it be Methanomicrobium? But this genus belongs to the class of Methanomicrobia and not to Methanobacteria. Please clarify this part of the MS!.

Line 264: Bacilla... Please correct it to Bacilli, and not just here, but throughout the manuscript!

Author Response

We are very grateful to Reviewer for their critical remarks which were read carefully and appropriate corrections have been made in the manuscript.

The authors have significantly improved the quality of the manuscript. However, some minor modifications are still needed.

In case of Figure 2 the assignment of the colours is incomplete! Class of Methanobacteria is missing in case of the class level diagram, while at least two phyla are missing in case of the phylum level diagram (red and dark blue phyla).

Answer: We agree with the reviewer’s remark. The Figure 2 has been corrected.

Line 260: "The Methanomicrobia genus..." Such genus does not exist! Should it be Methanomicrobium? But this genus belongs to the class of Methanomicrobia and not to Methanobacteria. Please clarify this part of the MS!.

Answer: We agree with the reviewer’s remark. The manuscript has been corrected. Line 260-262.

Line 264: Bacilla... Please correct it to Bacilli, and not just here, but throughout the manuscript!

Answer: We agree with the reviewer’s remark. The manuscript has been corrected. Line 265, 268.